# Structure-dynamics relationships in cryogenically deformed bulk metallic glass

Florian Spieckermann [1✉], Daniel Şopu [2,3], Viktor Soprunyuk[2,4], Michael B. Kerber[4], Jozef Bednarčík [5,6], Alexander Schökel [5], Amir Rezvan[2], Sergey Ketov[2], Baran Sarac [2], Erhard Schafler[4] & Jürgen Eckert[1,2]

The atomistic mechanisms occurring during the processes of aging and rejuvenation in glassy materials involve very small structural rearrangements that are extremely difficult to capture experimentally. Here we use in-situ X-ray diffraction to investigate the structural rearrangements during annealing from 77 K up to the crystallization temperature in $Cu_{44}Zr_{44}Al_8Hf_2Co_2$ bulk metallic glass rejuvenated by high pressure torsion performed at cryogenic temperatures and at room temperature. Using a measure of the configurational entropy calculated from the X-ray pair correlation function, the structural footprint of the deformation-induced rejuvenation in bulk metallic glass is revealed. With synchrotron radiation, temperature and time resolutions comparable to calorimetric experiments are possible. This opens hitherto unavailable experimental possibilities allowing to unambiguously correlate changes in atomic configuration and structure to calorimetrically observed signals and can attribute those to changes of the dynamic and vibrational relaxations ($\alpha$-, $\beta$- and $\gamma$-transition) in glassy materials. The results suggest that the structural footprint of the $\beta$-transition is related to entropic relaxation with characteristics of a first-order transition. Dynamic mechanical analysis data shows that in the range of the $\beta$-transition, non-reversible structural rearrangements are preferentially activated. The low-temperature $\gamma$-transition is mostly triggering reversible deformations and shows a change of slope in the entropic footprint suggesting second-order characteristics.

[1] Department of Materials Science, Chair of Materials Physics, Montanuniversität Leoben, Jahnstraße 12, 8700 Leoben, Austria. [2] Erich Schmid Institute of Materials Science of the Austrian Academy of Sciences, Jahnstraße 12, 8700 Leoben, Austria. [3] Institut für Materialwissenschaft, Fachgebiet Materialmodellierung, Technische Universität Darmstadt, Otto-Berndt-Strasse 3, Darmstadt D-64287, Germany. [4] Faculty of Physics, University of Vienna, Boltzmanngasse 5, 1090 Vienna, Austria. [5] Deutsches Elektronen Synchrotron (DESY), Notkestraße 85, 22607 Hamburg, Germany. [6] P. J. Šafarik University in Košice, Faculty of Science, Institute of Physics, Park Angelinum 9, 041 54 Košice, Slovakia. ✉email: florian.spieckermann@unileoben.ac.at

The atomistic mechanisms underlying the aging and rejuvenation of bulk metallic glasses (BMGs) still remain unclear to a great extent. The first studies on aging due to the mechanical degradation of glassy polymers emerging in the 1950s and the fact that aging and rejuvenation occur (as discussed by Kovacs[1]), culminated in a lively discussion about the existence of rejuvenation by Struik and McKenna in the late 1990s and the early years of this millennium[2,3]. As aging leads to enhanced brittleness, it was found to be detrimental for many potential applications of BMGs. Many efforts have been undertaken to tune the aging and rejuvenation. The degree of rejuvenation and, hence, the amount of the stored energy as well as the free volume in a BMG can be controlled by different methods such as deformation[4,5], high-pressure torsion (HPT)[6], ion irradiation[7], flash annealing[8], or even cooling to cryogenic temperatures[9,10]. All these studies reveal promising enhancement of mechanical properties without fully resolving their atomistic origins.

Recent developments in the understanding of the atomic-scale mechanisms of rejuvenation from computer simulations have developed very insightful knowledge regarding the structural and thermodynamic origin of aging and rejuvenation due to either thermal processing or mechanical (or cyclic) loadings[11–13]. The kinetic aspects, as well as the experimental proof of these theoretical findings, which relate the dynamic relaxation modes to the (structural) atomic-scale reorganization processes during ageing and rejuvenition in metallic glasses are however still sparse.

Different studies have tried to correlate the stress-driven processes, such as activation of shear transformation zones (STZs) and shear-banding, with thermally activated dynamic relaxations, i.e., $\beta$- and $\alpha$-relaxation modes[14–17]. The $\alpha$-relaxation is typically a low-frequency mode ($10^{-2}$ Hz) commonly associated with the glass transition, i.e., large cooperative rearrangements. The $\beta$-relaxation is a higher frequency mode ($10^3$ Hz) related to structural rearrangements on a smaller scale in the glassy state. The coupling of $\alpha$- and $\beta$-modes sometimes results in the observation of an excess wing in the loss modulus. More recent studies suggested a third dynamic relaxation mechanism, termed $\gamma$- or $\beta'$-relaxation, activated at low temperatures for low-frequency actuation. The formation and relaxation of stress inhomogeneities at cryogenic temperatures might be correlated to this relaxation mechanism[18,19]; however, little is known about the structural origin of this relaxation due to its recent discovery and the experimental challenges associated with cryogenic cooling.

The present paper aims to give deeper insight into the interplay of structural reorganization of the material and the dynamic relaxations. The understanding and experimental validation of the mechanisms occurring on the atomistic scale in glassy materials and particularly in bulk metallic glasses during aging and rejuvenation are crucial to improve and understand the origins of their limited ductility. Many structural characterization methods, however, fail to catch the very small changes related to aging and rejuvenation in metallic glasses. In this work, we use in situ synchrotron X-ray diffraction to study the structural rearrangements during annealing from 77 K up to the crystallization temperature of $Cu_{44}Zr_{44}Al_8Hf_2Co_2$ BMGs. This is done by determining small configurational changes in topological ordering with high time and temperature resolution. We propose here to use an equivalent of a configurational entropy of the experimentally determined X-ray pair distribution function (PDF) to make the subtle changes occurring during annealing visible. The samples were rejuvenated by high-pressure torsion (HPT) performed at cryogenic and room temperatures prior to the in situ annealing experiments. The as-deformed state of the samples was preserved by cryogenic storage prior to the in situ annealing experiments (please refer "Methods" section for details). Structural changes reflected in the X-ray-derived equivalent configurational entropy are correlated with dynamic mechanical analysis (DMA) as well as with differential scanning calorimetry (DSC) to determine dynamic relaxations and crystallization. The DMA measurements provide a clear picture of the relaxation process and are able to identify and distinguish between the well-known $\beta$- and $\alpha$-relaxation modes and also reveal the presence of the fast $\gamma$-relaxation mechanism in the glassy material.

## Results

**Structural characteristics.** Figure 1 shows selected reduced pair distribution functions (PDFs) $G(r)$ as determined by X-ray diffraction at 303.15 K while heating in situ. It can be seen that the differences are large for the as-cast state where no shoulder (see inset in Fig. 1c) is discernible in the first peak, whereas for the other two samples (deformation at RT and at 77 K) that were immediately stored at 77 K after HPT a clear shoulder is discernible. More intriguing is the fact, that the sample deformed by HPT at room temperature and relaxed for 7 days does not exhibit the shoulder and also the second peak (above 5 Å medium-range order (MRO) is considered to start) approximates the as-cast state and the two present shoulders smear out as well. Upon further heating, the shoulders also start to disappear for the cryogenically stored samples (Supplementary Movies 1 and 2). The shoulder is an indication for HPT-induced short-range ordering. This very local structure is related to the elevated pressure during HPT and is not stable at room temperature and ambient pressure.

The development of a shoulder in the first diffraction peak indicates that clusters are affected by the HPT deformation process in the short-range order (SRO) regime. BMGs have a high degree of SRO, and the clusters in their structure have a preference to develop fivefold symmetry (close-packed)[20]. The hydrostatic stress during the HPT process rejuvenates the glassy structure by increasing the free volume. At the same time, the amount of local ordering and the fraction of favored fivefold motifs[21,22] are increased and stabilized by hydrostatic stress[5]. This effect is not present in the sample stored at room temperature after HPT. Here, local reconfiguration via thermally induced rearrangement can relax the structure.

The entropy is a useful measure of the energetic state of glass concerning aging and rejuvenation[23]. Multibody entropies have been derived since the early stages of statistical physics. Baranyai and Evans showed that two-body contributions[24], the pair correlation, dominate the configurational entropy of a liquid. With low temperature resolution, the configurational entropy was used to study liquid-to-liquid phase transitions in $In_{20}Sn_{80}$[25]. Here, we use a measure of the configurational entropy as calculated from the pair correlation function for further analysis. The pair correlation function $g(r)$ can be calculated from the experimentally determined reduced pair distribution (correlation) function $G(r)$ by

$$g(r) = \frac{G(r)}{4\pi\rho_0 r} + 1 \qquad (1)$$

with $\rho_0$ the mean atomic number density of the alloy determined from the slope of $G(r)$ in the range between 0 and 2 Å and $r$ the radius.

The analysis of the measured $G(r)$ suggests that the correlational equivalent of the configurational entropy $S_{eq}$— derived from the two-body correlation—could be used as a measure of the state of aging or rejuvenation of the metallic glass, comparable to the approach applied by Piaggi et al.[26] on pair correlation data derived from molecular dynamics simulations. The configurational entropy $S_{eq}$ can then be calculated with the equation

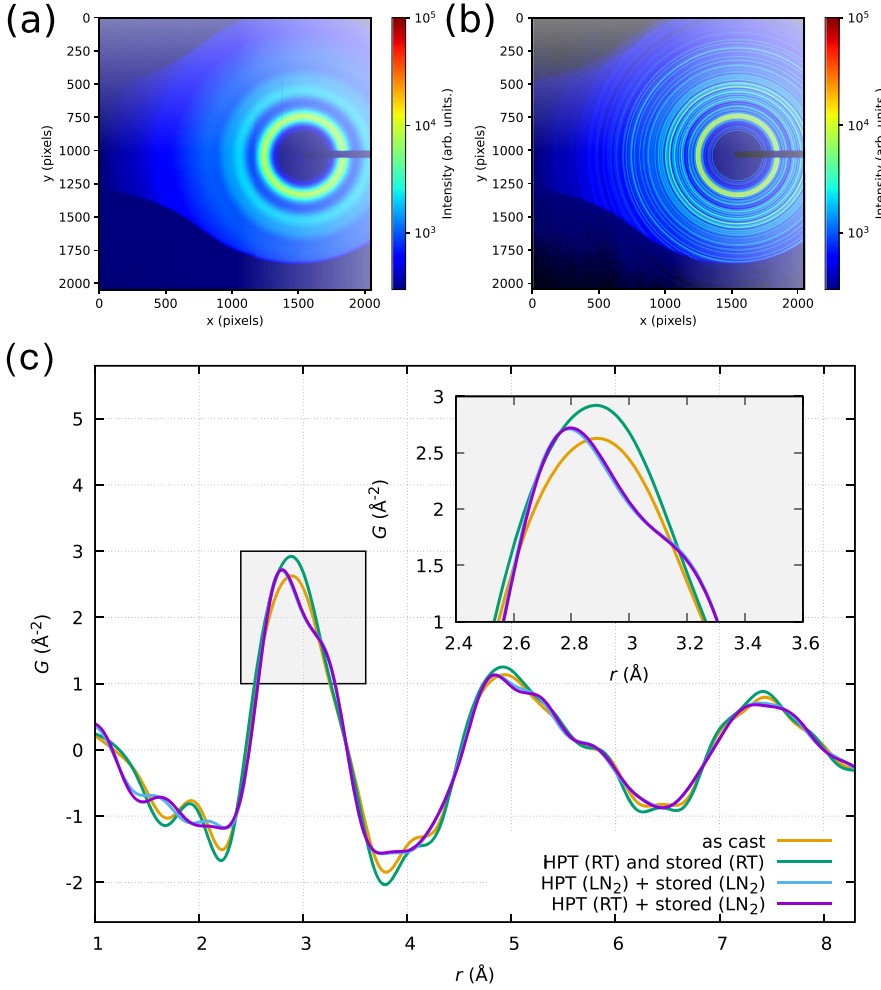

**Fig. 1 2-D diffraction and pair distribution function. a** Diffraction pattern of an as-cast amorphous sample, (**b**) diffraction pattern of the crystallized state. **c** Comparison of the reduced pair distribution function $G(r)$ at room temperature for the as-cast, and three different HPT-deformed states. If no cryogenic storage is ensured after HPT, the sample relaxes within 1 week and the shoulders in the first and second peaks of $G(r)$ smear out indicating a certain degree of disordering/aging.

derived by Nettleton and Green[27,28]

$$S_{eq} = -2\pi\rho_0 k_B \int \left( g(r)\ln g(r) - g(r) + 1 \right) r^2 dr, \qquad (2)$$

with $k_B$ the Boltzmann constant. It has been derived to determine the two-body contribution to the excess configurational entropy of a single-component liquid. The definition used here refers to the excess entropy with respect to the gas state[29] (other definitions sometimes used in glass physics refer to the excess with respect to the relaxed glass or to the crystalline state[23,30]).

We apply this formula to the experimentally derived XRD pair correlation function. As such, we treat the five-component metallic glass in a first approximation similar to a monoatomic liquid, which is the condition for which the present formula has been derived and tested. As the formula is nonlinear, the derived entropy can only be used to visualize structural differences in the same material but not to directly derive quantitative results. Such results would require the determination of partial PDF's, which is planned for future work. For small changes in the topological ordering which occur before and during glass transition, the evaluation does not become unstable. As crystallization involves also substantial chemical reordering, any derived entropies from the above formula should, however, be treated with great care in order to avoid unphysical conclusions.

For the calculation of the equivalent configurational entropy, it is assumed that the uncertainty introduced by the ad-hoc variational corrections applied by PDFgetX3[31] is small enough to assume a linear dependence.

We decided here not to derive the partial PDF data for the following reasons. The high number of components would make the derivation of the partial PDFs unreliable. We also aim at proposing an a priori model-free approach to assess structural disordering. In order to underline the difference of the quantity derived here from a correlational entropy (which would be the sum of all partial entropies), we will denote it as "equivalent entropy $S_{eq}$". Since contributions to the configurational entropy other than the two-body correlation (i.e., chemical) are not considered, the equivalent entropy is smaller than the total entropy.

Figure 2 depicts the resultant equivalent entropies as derived from the in situ diffraction experiments. The configurational state reflects the rejuvenation of the cryogenically stored samples with respect to the as-cast state. The HPT sample stored at room temperature relaxed into a state where $S_{eq}$ was closer to the as-cast state. Crystallization was characterized by a rapid reduction of $S_{eq}$ upon heating. After reaching crystallization, the samples were cooled to room temperature and a curve for the crystalline state was recorded. The glass transition $T_g$ can be seen at 709 K (in good agreement with the literature[32]).

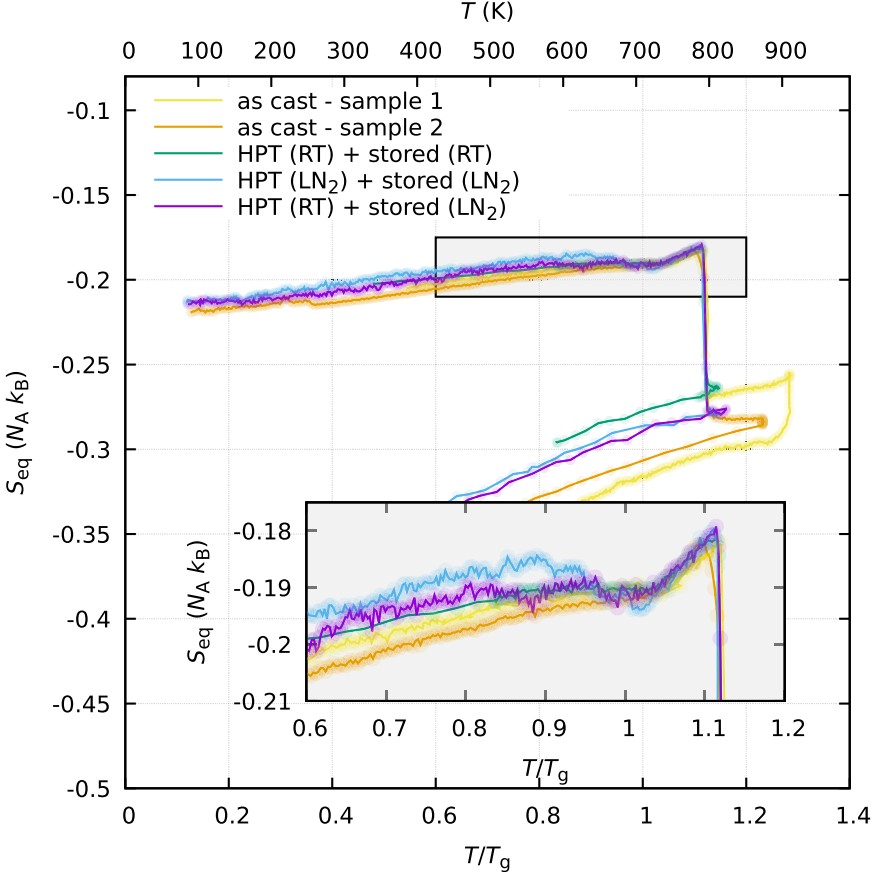

**Fig. 2 Change in equivalent configurational entropy as determined by Eq. (2).** Each point is derived from a reduced PDF calculated from an X-ray diffraction pattern. The curves have been shifted by an additive constant on the ordinate axis ($S_{eq}$-axis) in order to assure overlapping in the liquid state (unshifted data in Supplementary Fig. 1). The lower x axis is normalized to the glass transition temperature $T_g = 709$ K.

In the glassy state, characteristic changes of the slope of $S_{eq}$ occur which are correlated in Fig. 3a with the dynamic transitions. In thermodynamic equilibrium, the entropy can be used to calculate an equivalent configurational heat flow $\Delta\phi_{eq}$ for a given heating rate $\beta_h$ with the equation

$$\Delta\phi_{eq} = \frac{T \cdot dS_{eq}}{dT} \cdot \beta_h = \Delta c_{eq,p} \cdot \beta_h, \qquad (3)$$

where $T$ is the temperature and $\Delta c_{eq,p}$ the change in equivalent configurational heat capacity at constant pressure. By numerical differentiation (after smoothing) and application of Eq. (3), the equivalent configurational heat flow $\Delta\phi_{eq}$ as depicted in Fig. 3b can be correlated with the DMA-derived dynamical relaxations (Fig. 3c). Especially in the region of the $\beta$-relaxation and the excess wing, structural rearrangements are triggered that lead to a change of slope in $S_{eq}$. Figure 4 shows an excellent correlation between the DSC and the XRD-derived heat flows for the as-cast state. Interestingly, an enthalpic peak at low temperatures is observable in both evaluations that might originate from the relaxation of casting-induced stresses, triggered for instance by the mobilization due to the $\gamma$-transition. After the glass transition (at $T/T_g = 1$), the calorimetric heat flow shows a clear endothermic enthalpic peak that is not visible in the equivalent configurational heat flow $\phi_{eq}$. $\phi_{eq}$ also increases when entering the undercooled liquid until crystallization occurs. Oxidation leads to a slight curvature of the calorimetric signal at elevated temperatures.

**Relaxation kinetics**. The kinetics of the dynamic mechanical relaxations have been studied using ex situ dynamic mechanical

analysis. The results of DMA experiments performed on the as-cast and HPT-deformed material are represented in Fig. 5a–c with a focus on the $\beta$ and the $\alpha$-transition. In accordance with the earlier figures, the transition intervals are shown in colors derived from the torsion DMA experiment (represented reference in Fig. 5a). Further experiments focusing on the frequency dependence of the $\gamma$-transition studied at low temperatures are presented in Fig. 6. Figure 5a shows tan $\delta$ of the as-cast state and the HPT-deformed state from cryogenic temperature up to the crystallization temperature. Both states show evidence of the $\gamma$ and the $\beta$-transition. In the undeformed case, the $\beta$-transition appears as a well-separated peak, while in the deformed case a pronounced excess wing with a slight shoulder is formed. The excess wing is very pronounced for the HPT-deformed case and less for the as-cast material (Fig. 5a–c). This small wing is not suppressed when heating below or into the glass transition (Fig. 5a–c). The excess wing has been interpreted to be related to the coupling of $\beta$- and $\alpha$-relaxation modes[33]. Such a coupling could explain the strong structural aging to a lower entropy state before the glass transition, as shown in Fig. 3, as it allows a transition from local atomic rearrangements (STZ activation) to larger collective rearrangements. Only the full activation of the $\alpha$-modes and their percolation would then allow the glass transition and therefore the occurrence of homogeneous viscous flow. In the present experiments, the appearance of the $\beta$-transition peak in tan $\delta$ can be suppressed by heating just over this transition followed by subsequent cooling (Fig. 5b). When heating deep into the supercooled liquid (Fig. 5c), the $\beta$ peak is also undetectable, but the tan $\delta$ is slightly increased. The $\beta$-transition is considered to be related to the local plastic deformation (STZ activation) of metallic glasses.

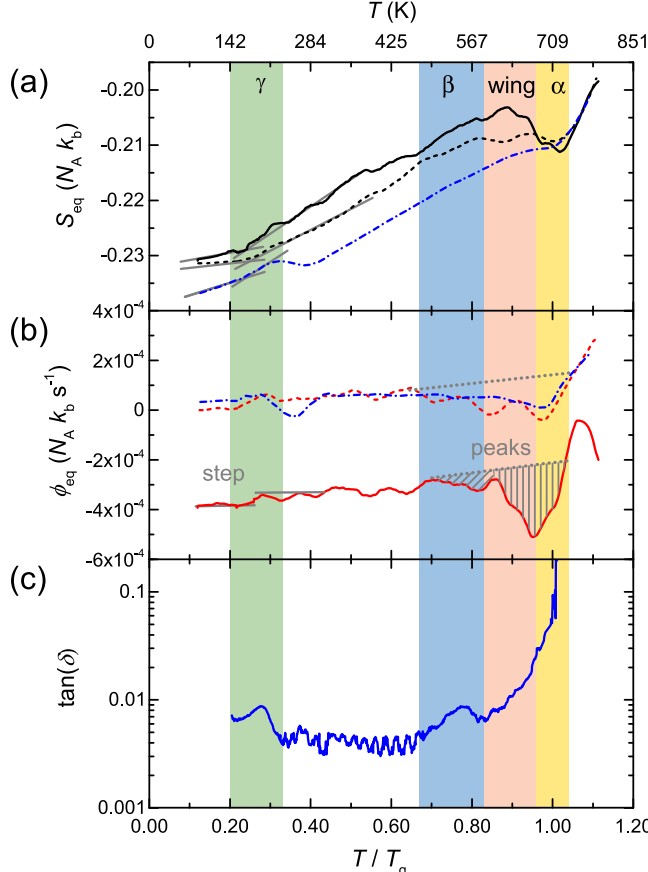

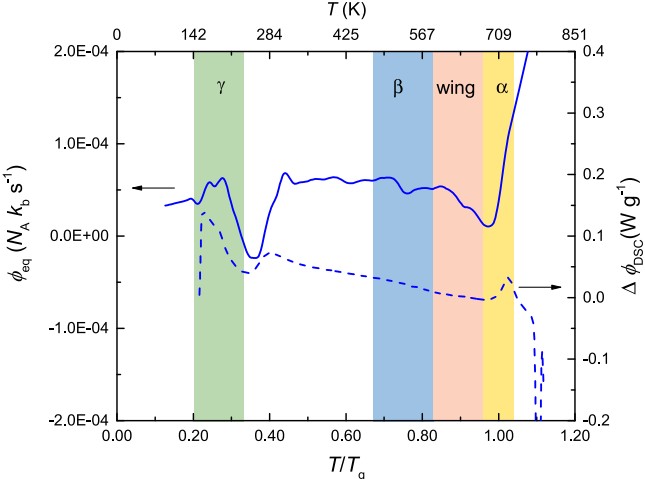

**Fig. 4 Calorimetric (dashed line) and equivalent configurational heat flow (solid line) for the as-cast state.** At low temperatures, a very clear peak is observable—in contrast to the HPT-deformed state.

**Fig. 3 Structure dynamics relationship.** Derivation of the equivalent configurational heat flow (shown in (**b**)) from the equivalent configurational entropy (shown in (**a**)). (dashed line) HPT at 77 K, (full line) HPT at RT, (dash-dotted line) as-cast. A change of slope at the $\gamma$-transition is indicated by solid gray lines in (**a**). Considerable enthalpic relaxation (the peaks are indicated by dashed gray lines in (**b**)) occurs when entering the $\beta$-relaxation region and between $\beta$ and $\alpha$ in the excess wing region. **c** Displays the loss tangent $\tan(\delta)$ as determined from torsion geometry DMA of the as-cast material.

For glasses rejuvenated by HPT or other methods (i.e., by affine cryogenic deformation, elastostatic loading, cold rolling, shot peening, uniaxial compression, triaxial compression) a fair amount of enthalpy relaxation is observed[4,6,9,34,35] in the temperature range characteristic for the $\beta$-transition. In the range of the $\beta$-transition recent literature reported that some glasses tend to stiffen in DMA[16,36] due to annihilation of free volume. It can hence be assumed, that rejuvenation and the $\beta$-transition are related.

In the present work, we observe that the occurrence of the $\beta$-transition peak is easily canceled by thermal treatment, indicating that the mobility of the underlying mobile species—i.e., deformation carriers or flow units—can be erased for our alloy. Considering that the local atomic mobility in the $\beta$-transition region is correlated with structural heterogeneities of enhanced free volume, this result can be considered as a further strong evidence for the viscoelastic and non-affine nature of the deformation mechanism associated with the $\beta$-transition.

Figure 6 shows DMA experiments carried out on an as-cast sample with different frequencies. The sample was cooled to 123.15 K and heated with 5 K/min and dynamic deformations with frequencies from 0.05 to 15 Hz have been applied on the same sample successively. Using Cole–Cole fitting, the transition temperatures were derived. Figure 7 shows the resulting Arrhenius evaluation that is used to determine the activation

energy of the $\gamma$-relaxation (Arrhenius evaluations for the $\alpha$- and $\beta$-relaxations are provided in Supplementary Fig. 2).

The $\gamma$-transition can be reproduced during each heating-cooling cycle. This confirms that the structural origin that allows this transition peak to be detectable is not destroyed when the experiment is performed without heating into the $\beta$-transition (which would allow annihilation of free volume, Fig. 6). These data suggest that the deformations active during the $\gamma$-transition are affine in nature but can be used to activate stress-driven non-affine relaxation/annihilation processes such as those reported by Ketov et al.[9]. The DMA-derived activation energies determined for the $\alpha$, $\beta$ and $\gamma$-relaxations are represented in Table 1.

All relaxation modes ($\alpha$, $\beta$, and $\gamma$) are the origin of aging (or rejuvenation) at different length- and timescales. This is reflected in the rather fast transition of the HPT-deformed sample stored at room temperature to an aged state featuring an entropy loss and a change in the peak shape of the first PDF maximum (SRO) and the second PDF maximum (MRO) in accordance with the results of Bian et al.[37] and Sarac et al.[38,39].

## Discussion

The DMA experiments which involve annealing in the $\beta$-relaxation regime and the supercooled liquid regime suppress the $\beta$-relaxation peak for subsequent experiments. Therefore, we interpret the $\beta$-relaxation and hence also the excess wing, to be of diffusional nature involving collective atomic movements/deformations (permanent). This also suggests that the $\beta$-relaxation is predominantly involved local, non-affine deformation. The excess wing may be interpreted as the activating mechanism allowing for the percolation of mobile species, hence confirming the nature of the wing as an overlap of $\alpha$- and $\beta$-modes. In contrast, the $\gamma$-relaxation is reproducible over many heating-cooling cycles (Fig. 6). This shows that the involved deformation is recoverable, suggesting predominantly affine recoverable structural rearrangements[40]. Although macroscopically BMGs are rather isotropic in nature, they are highly heterogeneous structures on the microscopic and atomic scales. Kinetic factors may furthermore allow a certain degree of super- or undercooling and, very recently, heterogeneity in time has been shown for glassy relaxations by means of X-ray photon correlation spectroscopy[41,42]. This temporal and structural heterogeneity is also reflected by broadened relaxation peaks due to deformation by HPT (Fig. 5a).

In the entropy curve, one may distinguish between a change of slope, corresponding to a first-order phase transition, and a

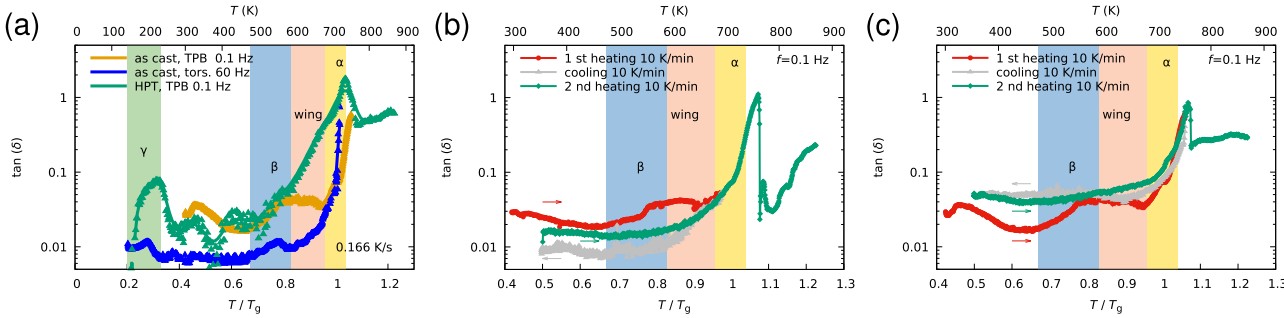

**Fig. 5 Kinetics of α-, β-, and γ-transitions. a** DMA Experiments on as-cast and HPT-deformed material. **b** β-transition upon heating just below $T_g$ and (**c**) upon heating deep into the undercooled liquid. In accordance with the previous figures, the transition intervals (shown in color) are derived from the torsion curve represented in (**a**).

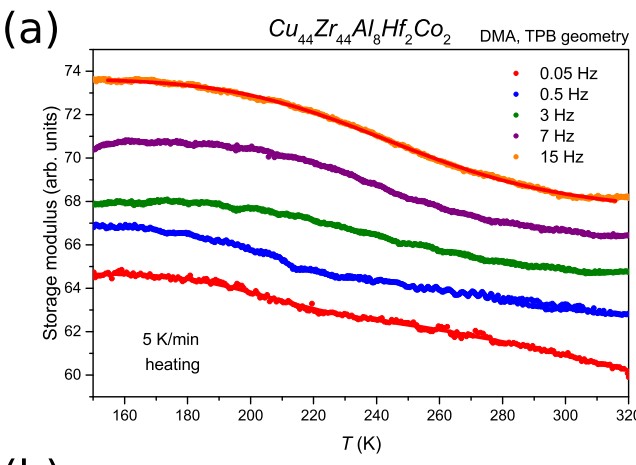

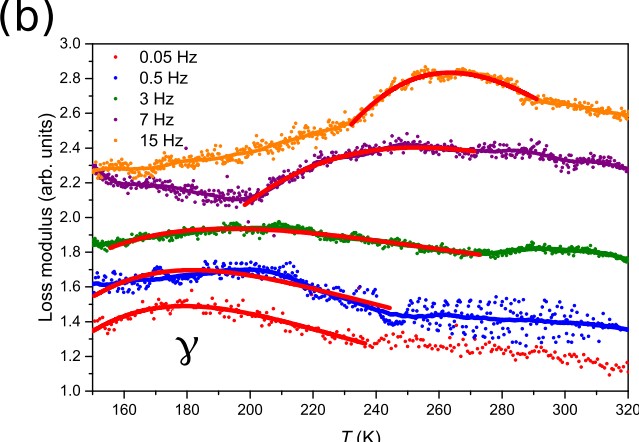

**Fig. 6 DMA experiments probing the γ-relaxation. a** Representing the storage modulus and (**b**) representing the loss modulus. Storage modulus curves for frequencies $f = 0.05, 0.5, 7,$ and 15 Hz are shifted on the ordinate axis from the 3 Hz data for clarity. Cole–Cole fits of the data indicated in (**b**) by red lines are used for the determination of the activation energy.

non-monotonous step for second-order transitions. Crystallization clearly involves a sharp step and can hence be interpreted as a first-order transition (Fig. 2). If the transition smears out over a larger temperature regime for instance due to structural effects (e.g., melting of crystals with different sizes), the separation becomes more difficult. Such behavior would result in a broadened peak in the heat flow. The percolated cooperative motion in the β-transition region can be seen as a thermal annealing process responsible for the annihilation of free volume. This is reflected by several overlapping peaks in the equivalent configurational heat flow. The peak in the equivalent configurational

heat flow may be an indication of overlapping processes with characteristics of a first-order phase transition. The excess wing as a result of coupled α- and β-modes seems to be related to a substantial peak of the equivalent configurational heat flow shown in Fig. 3 for the HPT-deformed state.

To our understanding, the HPT process increases the number of more mobile species (softer heterogeneities) in the glassy state. This is characterized by a transition to a higher entropy amorphous state of glassy nature that allows activating β-type string-like[43,44] and vortex-like motion[45–47], in contrast to liquid-like complete cooperative flow for the supercooled liquid state. This leads to the well-known phenomenon of strain localization in shear bands[48] during deformation processes. During the in situ heating process performed in this work, the glass will tend to relax to a lower entropy state using the thermodynamically and kinetically favorable β-relaxation modes (Fig. 8).

This could imply that a brittle-to-ductile transition involves the β-transition while the material is still in the solid state. The activation of mobile species through the β-transition can therefore be considered as a local bond-breaking mechanism with release of latent heat leading to an apparent character of a first-order phase transition in the equivalent configurational heat flow. The percolation of such localized events is necessary to achieve plastic flow in the metallic glass, schematically represented in the potential energy landscape in Fig. 8. In colloidal glasses, it has been reported that the failure transition may be associated with a first-order phase transition[49,50]. Evidently, the XRD-derived equivalent configurational entropy has high potential for giving new insights into the phenomena related to the glass-to-liquid transition[17,51–53].

This prompts the question whether the local bond-breaking mechanisms or structural rearrangements related to the β-transition in metallic glasses may be attributed to a first-order phase transition. Recent results reported in the literature interpret the β-transition as a mobilizing mechanism allowing for a polyamorphic, nucleation-controlled transition from the rejuvenated state to an aged state[48]. Over the last few years, the random first-order transition (RFOT) theory is one of the models for the glassy behavior that is able to well explain a number of phenomena observed in glassy materials with respect to their mechanical properties[54]. Recently, also the first-order characteristics have been shown for the glass transition of ultra-fragile glasses[55]. The observations made in the framework of this work are not conclusive in this respect; however, the equivalent configurational heat flow, as well as calorimetric heat flow, are associated with enthalpic peaks related to the β-transition that can be interpreted as showing first-order characteristics.

In summary, the pair distribution function was continuously measured to calculate the entropy of the pair correlation (excess equivalent configurational entropy $S_{eq}$) that allows us to derive an

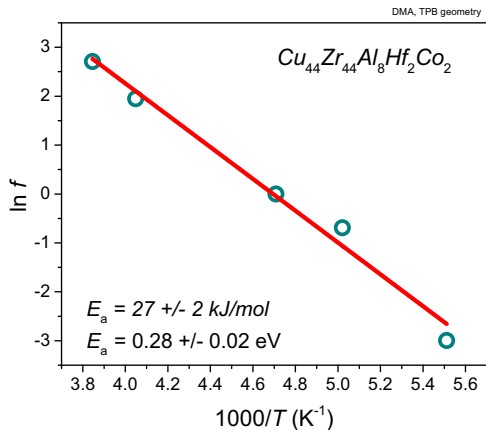

**Fig. 7 Activation energy of the γ-relaxation.** Arrhenius plot of the frequency ($f$) dependence of the $γ$-transition yielding activation energy on the order of $27 \pm 2$ kJ mol$^{-1}$ = $0.27 \pm 0.02$ eV.

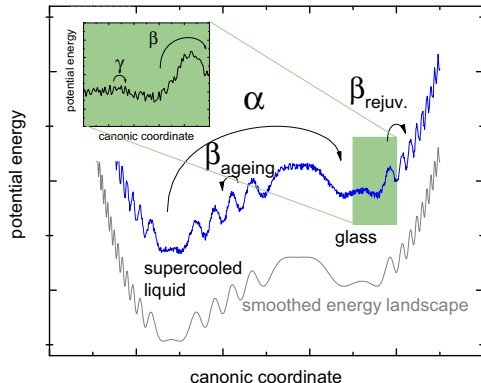

**Fig. 8 Potential energy landscape scheme.** Schematic representation of the energetic scales of the relevant dynamic relaxation modes at a constant temperature $T < T_g$.

**Table 1 Activation energies for the α-, β-, and γ-relaxation derived from the DMA experiments.**

| Relaxation | Activation energy $E_A$ (eV) |
| --- | --- |
| $α$ | $7.51 \pm 0.93$ |
| $β$ | $1.29 \pm 0.31$ |
| $γ$ | $0.28 \pm 0.02$ |

**Table 2 Sample matrix showing the number of HPT rotations, deformation temperature ($T_{def.}$) and storage temperature for all sample treatments $T_{stor.}$.**

| Label | Rotations | $T_{def.}$ (K) | $T_{stor.}$ (K) |
| --- | --- | --- | --- |
| As cast | 0 | - | 300 |
| HPT (RT) + stored (RT) | 20 | 300 | 300 |
| HPT (LN$_2$) + stored (LN$_2$) | 20 | 77 | 77 |
| HPT (RT) + stored (LN$_2$) | 20 | 300 | 77 |

equivalent configurational heat flow of the changes in pair correlation with unprecedented high-temperature resolution. We present here evidence that $S_{eq}$ changes in a characteristic manner when moving through the temperature regimes of the dynamic relaxations ($γ$, $β$, excess wing, and $α$). The structural rearrangements are occurring at the same temperatures as the dynamic $γ$- and $β$-relaxations, the excess wing as well as the $α$-relaxation determined by DMA. The $β$-transition and the excess wing have stronger first-order characteristics, as indicated by clear peaks in the equivalent configurational heat flow. These peaks may be related to local bond-breaking mechanisms leading to symmetry changes on the level of clusters. $S_{eq}$ as derived from the pair correlation shows that the $β$-transition, the excess wing, and the $α$-transition have very clear structural footprints. Our work also shows some indication that the $γ$-transition might be related to a change of slope of $S_{eq}$, characteristic for a second-order phase transition similar to the $α$-relaxation.

The present work suggests to use $S_{eq}$ as a measure for the state of rejuvenation of metallic glasses. With improving synchrotron sources, the possibility to derive configurational entropy data with high time or spacial resolution may spark new research to better understand the nature and structural origins of different relaxation mechanisms occurring in glassy alloys. In future works, the derivation of partial PDFs may even allow to quantitatively determine the configurational entropy as a material property during in situ diffraction experiments, potentially unveiling to date unknown phenomena in metastable (amorphous) solids and liquids.

## Methods

The bulk metallic glass $Cu_{44}Zr_{44}Al_8Hf_2Co_2$, was chosen based on the well-studied bulk metallic glass system $Cu_{44}Zr_{44}Al_8$ with minor additions of $Hf$ to further increase the glass-forming ability[56] and the addition of $Co$ in order to maximize the ability of the material to rejuvenate by moderate atomic distance shortening[57].

Samples of $Cu_{44}Zr_{44}Al_8Hf_2Co_2$ were obtained by the suction casting of rods (3 mm diameter) and plates (1 mm thickness) in an Edmund Bühler arc melter

under vacuum after multiple purging with $Ar$ gas and purification by $Ti$ getter. For the deformation experiments, discs with diameter $d = 8$ mm and height $h = 1$ mm were prepared by grinding and fine polishing from the as-cast plates.

HPT deformation was chosen in the present paper as it allows inducing a high degree of rejuvenation[6]. HPT was performed up to 20 revolutions at a pressure of 6 GPa on an HPT press (type WAK-01 Mark 1) machine with martensitic chromium steel anvils. Cooling during deformation using liquid nitrogen was applied to selected samples in order to study the effect of cryogenic deformation. For room temperature deformed samples, the samples were cooled to liquid nitrogen (LN$_2$) temperature (77 K) by submerging the whole anvil setup in LN$_2$ in order to suppress relaxation. Only then, was the pressure removed and the sample transferred at 77 K to a transportation dewar allowing to "freeze-in" the state post-HPT before unloading the pressure. The "frozen" samples were then transported to the synchrotron and transferred with the cryogenically arrested status (post-HPT) to the cold (77 K) heating stage. The different sample states are reproduced in Table 2. No intermediate heating before the start of the synchrotron experiment was allowed by this procedure for the states labeled as HPT (LN$_2$) + stored (LN$_2$) and HPT (RT) + stored (LN$_2$).

In situ X-ray diffraction was performed on the P02.1 Powder Diffraction and Total Scattering Beamline of PETRA III using a Perkin Elmer XRD1621 (200 μm × 200 μm pixel size) detector with a photon energy of 60 keV in transmission setup. The cold sample, still at liquid nitrogen temperature, was mounted to the sample stage of a *LINKAM THS 600* temperature controller using a custom-built sample environment[58,59]. To prohibit a temperature gradient between the sample and the instrument, the sample stage was pre-cooled to a temperature of $93.15 \pm 1$ K. Wide-Angle X-ray diffraction (WAXD) patterns of 12 s were then recorded while heating with 10 K/min from 93.15 K up to 873.15 K. After crystallization, the samples were again cooled to room temperature with 50 K/min. The diffraction patterns were carefully calibrated using a $CeO_2$ reference (NIST 674b) and the pyFAI software[60], and background subtraction was performed for the sample container. Pair distribution functions were determined using the software PDFgetX3[61]. The mean atomic number density was determined from $G(r)$ on the interval $0-2$ Å, where $G(r) = -4\pi r\rho_0$. The nominal mean atomic number density of the alloy as calculated from the atomic composition amounts to $\rho_0 = 5.38345 \cdot 10^{28}$ m$^{-3}$.

Dynamic mechanical analyses were performed on a TA Instruments Discovery Hybrid Rheometer DHR 3 in torsional and three-point bending (TPB) mode. The experiments were performed by heating with constant heating rates from 123.15 to 873.15 K with heating rates varying from 2 to 10 K/min. For the determination of the activation energies, the frequencies were varied from 0.05 to 15 Hz in TPB mode and up to 60 Hz in a torsional mode. The loss tangent was fitted with the Cole–Cole equation in order to derive the frequency dependence of the transition temperatures.

Differential scanning calorimetry was performed on a Mettler Toledo DSC 3+ using platinum crucibles in a temperature range from 153.15 to 793.15 K with a heating rate of 20 K/min. The second heating signal was not subtracted.

## Data availability

The raw/processed data required to reproduce these findings can be provided by the corresponding author upon reasonable request.

## Code availability

The GnuOctave code required to calculate the equivalent entropies can be provided by the corresponding author upon reasonable request.

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

## Acknowledgements

We thank DESY (Hamburg, Germany), a member of the Helmholtz Association (HGF), for providing experimental facilities. Parts of this research were carried out at PETRA III using the Powder Diffraction and Total Scattering beamline P02.1. The research leading to our findings took place in the framework of project CALIPSOplus under the Grant Agreement 730872 of the EU Framework Programme for Research and Innovation HORIZON 2020 (F.S., B.S., and E.S.). This work was funded by the European Research Council under the ERC Advanced Grant INTELHYB (grant ERC-2013-ADG-340025, J.E., B.S., and A.R.), the ERC Proof of Concept Grant TriboMetGlass (grant ERC-2019-PoC-862485, J.E.), and by the Austrian Science Fund (FWF) under project grant I3937-N36 (B.S. and A.R.). We thank Dr. Christoph Gammer and Dr. Ivan Kaban for fruitful discussions. Cameron Quick, MChem. is thanked for English language editing.

## Author contributions

F.S., E.S., B.S., and J.E. designed the research. F.S., E.S., B.S., M.K., J.B., and A.S. performed the synchrotron experiments. F.S. and D.S. evaluated and cured the data. V.S. performed the DMA experiments. F.S. performed the DSC experiments. S.K., B.S., and A.R. produced the alloy. F.S. wrote the original draft. J.E. and B.S. provided funding. All authors contributed to the interpretation of the data and revised the manuscript.

## Competing interests

The authors declare no competing interests.
