## [Peer Review File · Nature Communications]

REVIEWER COMMENTS

Reviewer #1 (Remarks to the Author):

This manuscript reports synchrotron X-ray diffraction characterization of the a CuZrAlHfCo bulk metallic glass under high pressure torsion at both room temperature and 77 K. The measurement allows determination of the pair distribution function at different temperatures, based on which a two-body configurational entropy, termed excess equivalent configurational entropy, is further proposed and calculated. After that, a novel equivalent configurational heat flow can be defined. It is very interesting to find that the latter derived from the two-body excess entropy (RDF) is in accordance with direct DSC evaluation, the calorimetric heat flow. The heat flow is also compared with dynamical mechanical relaxation, which allows detailed discussion on the link between relaxation modes (α , β , γ relaxations) and the structural change in terms of configurational entropy and heat flow. And the connection between relaxation and glass transition is also touched.

In principle, this is a novel work with new insights from direct XRD characterization. The analyses are sophisticated and the discussion is comprehensive. The conclusions are basically supported by the experimental data. The findings are interesting to the broad community of glass physics and materials sciences. The paper is well-organized and clearly written. In sum, the scientific merit and high impact has almost justified publication in Nature Communication. However, I would like to bring about a list of suggestions to further improve the scientific rigor and representation of the manuscript. The comments and suggestions are listed below.

The schematic plot of potential energy landscape in Fig. 8 is misleading. It is well-known that the potential energy of liquid is higher than that of glass. Therefore, the relative heights of different phases should be updated.

This manuscript misses most of the recent developments in understanding of the atomic-scale mechanisms of rejuvenation from computer simulations, which have developed very insightful knowledge to the structural and thermodynamical origin of rejuvenation due to either thermal processing or mechanical (or cyclic loadings) loadings. It is already revealed that cavitation induced

The introduction starts with aging and rejuvenation, but suddenly transitions to discussion of different relaxation modes (dynamical mechanical relaxation) without connect the two points. It seems the motivation of the present work is not well documented in the introduction section. What the real question is and where the difficulty lies should be explained clearly, even to the potential readers without expertise in metallic glasses or amorphous materials.

In Fig. 2, the most relevant data domain is those below the overlapping in the liquid state. It is therefore to magnify the section, e.g. by providing an inset of the main figure.

In page 4, line 16, it is claimed "...rejuvenates the glassy structure by increasing the free volume". However, one can notice from the $G(r)$ shown in Fig. 1(c) that intensity of first peak increases due to HPT, and structural ordering occurs. How to understand the two seemingly conflicting observations? Is that due to the critical roles of medium-to-long range order of metallic glass?

It is not clear what " Δ " means in configurational entropy, as defined by Eq. (2). In fact, this definition indicates the reduction of two-body component of configurational entropy compared to an ideal gas state due to positional correlation. In the community of glass physics, excess entropy is usually defined as the liquid entropy over the glass entropy. The two concepts of entropy should not be confused.

Since the chemical compositions are not distinguished (although in principle it can be done) in the definition of the equivalent configurational entropy (with only two-body contribution), it should be noted that the present number of entropy is much underestimated.

It is not intuitive that the magnitude of equivalent configurational heat flow of the HPT sample at RT is much lower than others, as reported in Fig. 3b.

The judgement of nonaffine nature of β relaxation, and affine nature of γ is rather speculative. Further evidences from either experiments or simulations are necessitated before reaching a solid conclusion. To be honest, I agree that this is a very difficult task, but the authors should mention it somehow since the present mean-field sense characterizations do not allow direct quantitative description of affine/nonaffine deformation at atomic scale, e.g. down to the level of the well-known nonaffine squared displacement proposed by Falk and Langer.

It is recently proposed Shannon entropy about distribution of local atomic structures as an effective metric of the configurational entropy that drives the glass transition in metallic glass-forming liquid [Han et al., Atomistic structural mechanism for the glass transition: Entropic contribution, Phys. Rev. B 101, 014113 (2020)], which supports the present views of structural ordering/disordering effect on the aging and rejuvenation of metallic glasses.

Typos: middle of page 3 "153,15 K" should be "153.15 K".

Grammar mistakes: in page 4, "...five-fold motifs is increases ...".

Reviewer #2 (Remarks to the Author):

The paper "Structure-dynamics relationships in cryogenically deformed bulk metallic glass" by Spieckermann et al. presents a comprehensive study of structural transformations in a rejuvenated bulk metallic glass. The main innovation of this work is the calculation of the configurational entropy from x-ray diffraction (XRD) data with high resolution in temperature and time. The results are also compared with data from other methods such as dynamic mechanical analysis and differential scanning calorimetry. The authors find that the beta transformation is non-reversible while the gamma transition is reversible. They also claim that they find evidence consistent with a first order phase transformation for the beta-transformation and a second order phase transition for the gamma-transformation.

The approach presented by the authors based on the calculation of the entropy from x-ray diffraction data provides useful information and I think it may influence the community working on phase transformations. This is a very interesting paper, yet the quality of the presentation is not particularly good and I had to make an effort to understand the reasoning of the authors. To be more precise, it is not clear in the discussion of the results what has been found in the literature and what the authors infer from their experiments. Furthermore, I have serious doubts about the ability of the data to support the conclusions. In particular, I do not see evidence to support the claim about the order of the phase transformations.

In my opinion this might be a paper suitable for Nature Communications but the quality of the presentation has to be improved and the authors should clarify in the manuscript the points that I discuss in detail below.

1) I think Eq. (1) is missing an r in the denominator. See for instance this link <https://journals.iucr.org/j/issues/2014/03/00/kk5148/kk5148sup1.pdf>. The missing r makes the correct equation satisfy the important property that $g(r)=0$ at short distances when $G(r)=-4 \pi r \rho_0$. It is of the utmost importance to clarify if the authors used the correct expression for $g(r)$ to calculate the pair entropy. Otherwise all figures and analysis have to be done again.

2) In Fig 2, did the authors shift the curves by an additive constant or scaled them by a factor? Please clarify in the caption. Also, it would be good to add the value of the glass transition temperature in the caption. I also don't understand why shifting the curves was necessary. Please explain and see my next point.

3) I do not understand why arbitrary units are used for the entropy. The discussion about the corrections applied by PDFgetX3 does not seem to justify neglecting units. Please justify better why you used arbitrary units or use units of entropy, for example kJ/mol/K.

4) What is the maximum distance used in the calculation of the equivalent configurational entropy (upper limit of integration in Eq. 2)? Is it limited by the XRD data? Does it affect the results?

5) In page 5 the glass transition temperature is reported and it is not clear if it is from literature or calculated in this work. Add a citation if appropriate.

6) Fig 3. How were the intervals for the transitions (shown in colors) determined?

7) The discussion in section III. B. is obscure and hard to read. It jumps from Figure 6 to Figure 5. Then Figure 5 (a-c) is mentioned and it is not clear what the reader should focus on. It was also hard to understand what are results from the literature, what is found in this work, and what is the connection between the two. I think that improving this discussion would benefit the manuscript. Please check throughout the manuscript if results from the literature and what is found in this work is well separated and clear.

8) In Figure 5 a I do not understand why the authors included the blue curve "as cast, tors 60 Hz". I think it is not discussed and makes the figure harder to read. Did I miss the discussion? Also, Figures 5b and 5c would be easier to interpret if the regions of the transformation were colored as in 5a.

9) In Table II results for the activation energy of the alpha and beta transitions are reported. However, the data used for the calculation is not shown. Please report it in a supplementary information.

10) In page 9 a peak in the equivalent configurational heat flow is used as a justification to characterize the beta transitions as first order. Which peak are the authors referring to? Furthermore, what do the authors expect to see for first and second order phase transitions? A first order phase transition is characterized by discontinuous first derivatives of the free energy, i.e. the entropy and the volume. Did the authors find a discontinuous jump in the entropy?

11) The discussion in page 9 links the findings of this work with microscopic theories of deformation and relaxation. Please word the sentences in such a way that fact is differentiated from speculation.

12) The abstract states that "The results confirm that the structural footprint of the beta-transition is related to entropic relaxation with characteristics of a first-order transition." I do not see evidence in this paper to allow the use of the word "confirm". Please change it to "suggest" unless you can provide very strong evidence for a first order phase transition. Furthermore, what does entropic relaxation mean? Do the authors refer to a decrease in entropy during heating?

Answers to REVIEWER COMMENTS (in red color):

Reviewer #1 (Remarks to the Author):

This manuscript reports synchrotron X-ray diffraction characterization of the a CuZrAlHfCo bulk metallic glass under high pressure torsion at both room temperature and 77 K. The measurement allows determination of the pair distribution function at different temperatures, based on which a two-body configurational entropy, termed excess equivalent configurational entropy, is further proposed and calculated. After that, a novel equivalent configurational heat flow can be defined. It is very interesting to find that the latter derived from the two-body excess entropy (RDF) is in accordance with direct DSC evaluation, the calorimetric heat flow. The heat flow is also compared with dynamical mechanical relaxation, which allows detailed discussion on the link between relaxation modes (α , β , γ relaxations) and the structural change in terms of configurational entropy and heat flow. And the connection between relaxation and glass transition is also touched.

In principle, this is a novel work with new insights from direct XRD characterization. The analyses are sophisticated and the discussion is comprehensive. The conclusions are basically supported by the experimental data. The findings are interesting to the broad community of glass physics and materials sciences. The paper is well-organized and clearly written. In sum, the scientific merit and high impact has almost justified publication in Nature Communication. However, I would like to bring about a list of suggestions to further improve the scientific rigor and representation of the manuscript. The comments and suggestions are listed below.

We would like to thank the editors and the reviewers for finding the time and effort to consider our work and the positive appreciation. Both reviewers have requested substantial corrections to the introduction and to the discussion which we have addressed. In order to keep the answers to the revision requests clear we would like to refer to the differences file (diff.pdf) where these corrections are highlighted in color (additional text in blue and deleted text in red color).

The schematic plot of potential energy landscape in Fig. 8 is misleading. It is well-known that the potential energy of liquid is higher than that of glass. Therefore, the relative heights of different phases should be updated.

This is a good observation and the comment is true as there is a mistake in the scheme. The corrected figure contains the nomenclature "supercooled liquid". Below T_g the energy of the supercooled liquid is lower than the glassy energy. This is why the glass is able to relax towards a more relaxed "liquidlike" state. It should be noted that it can be distinguished between supercooling below different transition temperatures such as crystallization but also the glass-transition.

This manuscript misses most of the recent developments in understanding of the atomic-scale mechanisms of rejuvenation from computer simulations, which have developed very insightful knowledge to the structural and thermodynamical origin of rejuvenation due to either thermal processing or mechanical (or cyclic loadings) loadings. It is already revealed that cavitation induced

We thank Reviewer 1 for pointing out the missing the explanations in this respect. We included an extended comment in the introduction to address this deficiency and referred to several relevant publications. Please refer to the differences file (diff.pdf) to track these substantial changes.

The introduction starts with aging and rejuvenation, but suddenly transitions to discussion of different relaxation modes (dynamical mechanical relaxation) without connect the two points. It seems the motivation of the present work is not well documented in the introduction section. What the real question is and where the difficulty lies should be explained clearly, even to the potential readers without expertise in metallic glasses or amorphous materials.

Aging and rejuvenation are kinetic processes that involve the transition from a metastable state to a lower or higher energy state. Intuitively aging is involving an exoenergetic (exoentropic) step while rejuvenation is endoenergetic (endoentropic). The energy transfer could for instance be provided through heat leading to the known exothermic/endothemic signals in calorimetry for aging/rejuvenation.

The main question that is still experimentally unanswered for metallic glasses (in contrast to colloidal glasses and computer simulations) is the nature of the atomistic mechanisms driving aging and rejuvenation. Recent literature has given very clear indications that the dynamic relaxation mechanisms are related to atomic rearrangements and that these mechanisms also allow for the structural restructuring that is at the origin of aging and rejuvenation. The present work tries to link the dynamic mechanisms and the structural footprint of the atomic configuration.

While we cannot answer the question on the nature of the reorganization mechanism we believe that through the entropy evaluation of the PDF that we propose we have found a useful tool to correlate dynamic processes with structural reorganization.

We have revised the introduction in accordance with the above explanation in order to better highlight the main question. Please refer to the file "diff.pdf" to track the changes.

In Fig. 2, the most relevant data domain is those below the overlapping in the liquid state. It is therefore to magnify the section, e.g. by providing an inset of the main figure.

We magnified the overlapping section and thank Reviewer 1 for the useful suggestion.

Fig 1: Revised Fig.2 of the manuscript including the magnified region.

In page 4, line 16, it is claimed “...rejuvenates the glassy structure by increasing the free volume”. However, one can notice from the $G(r)$ shown in Fig. 1(c) that intensity of first peak increases due to HPT, and structural ordering occurs. How to understand the two seemingly conflicting observations? Is that due to the critical roles of medium-to-long range order of metallic glass?

The authors believe that more research will be necessary to ultimately answer the question at which point the free volume is formed during the HPT process. Our data indicates the free volume is formed by release of internal pressure/stresses during unloading. When the samples are stored at cryogenic temperature the state of elevated pressure is pertained and free volume is not (yet) formed.

It is not clear what “ Δ ” means in configurational entropy, as defined by Eq. (2). In fact, this definition indicates the reduction of two-body component of configurational entropy compared to an ideal gas state due to positional correlation. In the community of glass physics, excess entropy is usually defined as the liquid entropy over the glass entropy. The two concepts of entropy should not be confused.

We thank Reviewer 1 for indicating an inconsistent nomenclature in the literature. In the present paper we use the definition of Nettleton and Green that is referring to the difference with respect to the gas. We have made a clarification in the related section on page 5:

“ It has been derived to determine the 2-body contribution to the excess configurational entropy of a single component liquid. The definition used here refers to the excess entropy with respect to the gas state [35] (other definitions sometimes used in glass physics refer to the excess with respect to the relaxed glass or to the crystalline state [29,36]).”

We have also decided to omit the “\Delta”.

Since the chemical compositions are not distinguished (although in principle it can be done) in the definition of the equivalent configurational entropy (with only two-body contribution), it should be noted that the present number of entropy is much underestimated.

We have included a comment to point out this underestimation on page 5:

“Since contributions to the configurational entropy (i.e. chemical) other than the two body correlation are not considered the equivalent entropy is smaller than the total entropy.”

It is not intuitive that the magnitude of equivalent configurational heat flow of the HPT sample at RT is much lower than others, as reported in Fig. 3b.

To date, unfortunately, we have no good explanation for this difference and plan to perform further experiments. It might be related to a baseline problem.

The judgement of nonaffine nature of β relaxation, and affine nature of γ is rather speculative. Further evidences from either experiments or simulations are necessitated before reaching a solid conclusion. To be honest, I agree that this is a very difficult task, but the authors should mention it somehow since the present mean-field sense characterizations do not allow direct quantitative description of affine/nonaffine deformation at atomic scale, e.g. down to the level of the well-known nonaffine squared displacement proposed by Falk and Langer.

We understand that this is one of the big questions to be solved in glass physics. Experimentally the assessment of the atomistic differentiation between affine and non-affine deformations are still an unsolved challenge. Bubble raft experiments as well as computer simulations are giving some indications but miss the complexity of the atomic bonding in metallic glasses to a certain extent. In our discussion we can only refer to the integral data gained from XRD and DMA that indirectly provide some indications on the atomic processes. To our understanding the reversibility of a process is a good indication for the (globally) affine nature of the underlying process.

It is recently proposed Shannon entropy about distribution of local atomic structures as an effective metric of the configurational entropy that drives the glass transition in metallic glass-forming liquid [Han et al., Atomistic structural mechanism for the glass transition: Entropic contribution, Phys. Rev. B 101, 014113 (2020)], which supports the present views of structural ordering/disordering effect on the aging and rejuvenation of metallic glasses.

We thank Reviewer 1 for pointing out this interesting publication and have included the reference in the manuscript.

Typos: middle of page 3 “153,15 K” should be “153.15 K”.

Grammar mistakes: in page 4, “...five-fold motifs is increases ...”.

We thank the Reviewer for kindly pointing out these mistakes – they have been corrected.

Reviewer #2 (Remarks to the Author):

The paper "Structure-dynamics relationships in cryogenically deformed bulk metallic glass" by Spieckermann et al. presents a comprehensive study of structural transformations in a rejuvenated bulk metallic glass. The main innovation of this work is the calculation of the configurational entropy from x-ray diffraction (XRD) data with high resolution in temperature and time. The results are also compared with data from other methods such as dynamic mechanical analysis and differential scanning calorimetry. The authors find that the beta transformation is non-reversible while the gamma transition is reversible. They also claim that they find evidence consistent with a first order phase transformation for the beta-transformation and a second order phase transition for the gamma-transformation.

The approach presented by the authors based on the calculation of the entropy from x-ray diffraction data provides useful information and I think it may influence the community working on phase transformations. This is a very interesting paper, yet the quality of the presentation is not particularly good and I had to make an effort to understand the reasoning of the authors. To be more precise, it is not clear in the discussion of the results what has been found in the literature and what the authors infer from their experiments. Furthermore, I have serious doubts about the ability of the data to support the conclusions. In particular, I do not see evidence to support the claim about the order of the phase transformations.

In my opinion this might be a paper suitable for Nature Communications but the quality of the presentation has to be improved and the authors should clarify in the manuscript the points that I discuss in detail below.

We thank the Reviewer for his time in reviewing this article and positive appreciation.

1) I think Eq. (1) is missing an r in the denominator. See for instance this link <https://journals.iucr.org/j/issues/2014/03/00/kk5148/kk5148sup1.pdf>. The missing r makes the correct equation satisfy the important property that $g(r)=0$ at short distances when $G(r)=-4 \pi r \rho_0$. It is of the utmost importance to clarify if the authors used the correct expression for $g(r)$ to calculate the pair entropy. Otherwise all figures and analysis have to be done again.

We thank Reviewer 2 for carefully revisiting the mathematics. We found it was a mistake in the manuscript that was not present in the evaluation scripts (GnuOctave). The presented figures are correct.

2) In Fig 2, did the authors shift the curves by an additive constant or scaled them by a factor? Please clarify in the caption. Also, it would be good to add the value of the glass transition temperature in the caption. I also don't understand why shifting the curves was necessary. Please explain and see my next point.

We shifted the curves by an additive constant. We have also added the non-shifted curves in the supplementary information. According to the reviewers comment we added the glass transition temperature in the figure caption. The shifting was necessary to make the representation legible in terms of the relaxation mechanisms.

Fig 2: Revised Fig2 without shifting factor as included in the supplementary information(SI)

3) I do not understand why arbitrary units are used for the entropy. The discussion about the corrections applied by PDFgetX3 does not seem to justify neglecting units. Please justify better why you used arbitrary units or use units of entropy, for example kJ/mol/K.

We resorted to this nomenclature because the entropy calculated largely underestimates the total entropy. Also the calculated entropy is in relation to the gas state. In accordance with the Reviewers suggestion we have now resorted to units of $N_A k_b$ for the entropy.

4) What is the maximum distance used in the calculation of the equivalent configurational entropy (upper limit of integration in Eq. 2)? Is it limited by the XRD data? Does it affect the results?

Yes, we limited it with the XRD data. The impact on the entropy might come from the Fourier “wobbles” in $G(r)$ due to small r ranges.

5) In page 5 the glass transition temperature is reported and it is not clear if it is from literature or calculated in this work. Add a citation if appropriate.

We measured the glass transition in DSC. It is in excellent agreement with the XRD derived value as well as with literature [38].

6) Fig 3. How were the intervals for the transitions (shown in colors) determined?

The intervals were derived from the $\tan \delta$ curves derived at 60 Hz torsion mode with a heating rate of 10 K / min. The same heating rate was applied in the XRD experiments.

7) The discussion in section III. B. is obscure and hard to read. It jumps from Figure 6 to Figure 5. Then Figure 5 (a-c) is mentioned and it is not clear what the reader should focus on. It was also hard to understand what are results from the literature, what is found in this work, and what is the connection between the two. I think that improving this discussion would benefit the manuscript. Please check throughout the manuscript if results from the literature and what is found in this work is well separated and clear.

We would like to thank the Reviewer for this important point. The complexity of underlying processes is hard to summarize. We have attempted to better organize the section III B and have included a section III. C to make more clear what is derived from our experimental data (III. A. and III. C.) and where we link our findings to the recent literature (III. C.).

Please refer to the differences file (diff.pdf) for these substantial changes.

8) In Figure 5 a I do not understand why the authors included the blue curve "as cast, tors 60 Hz". I think it is not discussed and makes the figure harder to read. Did I miss the discussion? Also, Figures 5b and 5c would be easier to interpret if the regions of the transformation were colored as in 5a.

We have included the torsion curve to show that it is reproducible and a general phenomenon. It is also used as a reference for the color coding. We have added a description in the text and in the figure caption. We have also added the coloring in figure 5b and c

9) In Table II results for the activation energy of the alpha and beta transitions are reported. However, the data used for the calculation is not shown. Please report it in a supplementary information.

A related figure has been added as supplementary information.

10) In page 9 a peak in the equivalent configurational heat flow is used as a justification to characterize the beta transitions as first order. Which peak are the authors referring to? Furthermore, what do the authors expect to see for first and second order phase transitions? A first order phase transition is characterized by discontinuous first derivatives of the free energy, i.e. the entropy and the volume. Did the authors find a discontinuous jump in the entropy?

There is a smeared out step. This can be interpreted as a hint for such a behavior. We have further weakened our claim and have underlined that this is so far speculative but based on strong experimental indications.

11) The discussion in page 9 links the findings of this work with microscopic theories of deformation and relaxation. Please word the sentences in such a way that fact is differentiated from speculation.

We have substantially rewritten the section and separated the results and the facts from speculative discussions. We made very clear now where more research is necessary and formulated open questions that is sparked by our research. (sections IIIb and IIIc in the revised manuscript). Please refer to the differences file (diff.pdf) to track these changes.

12) The abstract states that "The results confirm that the structural footprint of the beta-transition is related to entropic relaxation with characteristics of a first-order transition." I do not see evidence in this paper to allow the use of the word "confirm". Please change it to "suggest" unless you can provide very strong evidence for a first order phase transition. Furthermore, what does entropic relaxation mean? Do the authors refer to a decrease in entropy during heating?

We thank the reviewer for this good suggestions. We have revised the manuscript and the abstract accordingly.

Entropic relaxation means a deviation of the entropy curve from the glassy behavior towards the supercooled (supercooled with respect to the fictive temperature) liquid line.

REVIEWER COMMENTS

Reviewer #1 (Remarks to the Author):

As I stated in my first report, this is a novel work which proposed a feasible solution for determination of (partial) configurational entropy of glass-forming system via X-ray pair correlation function. The strategy allows comprehensive discussions on important issues in glass physics, e.g., the aging/rejuvenation, relaxation, glass transition as well as atomic-scale mechanism for structural arrangement. All of which can be correlated with macroscopic calorimetric measurement. The scientific merit has justified publication in Nature Communications. It will definitely bring extensive attentions from the community, not only from experimentalists, but also those researchers in the computer simulations.

This is also a very good revision. The revised manuscript is now much solid, clear and errorless. All my concerns have been appropriately addressed. The explanations are reasonable and clear to me. Indeed it is a good demonstration of the peer review system. The revision is accepted.

With these, I would like to recommend publication of this revised manuscript as is.

Reviewer #2 (Remarks to the Author):

I thank the authors for carefully reading my comments and correcting the manuscript where appropriate. I believe that the discussion in section III C is now better structured and more informative.

I still have one concern about this manuscript. In my opinion there is still an issue with the discussion about the order of the beta-transition that was already highlighted in my first review. In an ideal first-order phase transition one would expect to see a discontinuous jump in the entropy and a clear peak in the heat flow. This is of course not observed in Fig 3 b and instead there is a relatively broad peak from $T/T_g \sim 0.85$ to $T/T_g \sim 1.05$ for the HPT at RT sample. I agree that this is compatible with a phase transformation with first-order characteristics. My only concern is that it is localized completely in the wing and alpha regions. Why are the authors ascribing the first order characteristic to the beta-transformation if the peak is located in the wing and alpha regions?

If the authors have a reasonable answer to this question, I think the paper should be published in its current form.

Minor points:

- 1) Fig 2. Caption. Replace "shifted by a scalar factor" by "shifted by an additive constant"
- 2) Section III A. "Baranyai and Evens ..." should be "Baranyai and Evans ..."
- 3) Section III C. "Crystallization clearly involves a sharp step and can hence be interpreted as a second order transition (Figure 2)" Please change "second order" to "first order". Of course crystallization is a first order transition.

Answers to REVIEWER COMMENTS (in red color):

We would like to thank the editors and the reviewers for finding the time and effort to consider our work and the positive appreciation. Reviewer 2 has requested further clarification which we address below. We have also corrected minor mistakes in the manuscript and adapted to the editorial requirements. For these changes we would like to refer to the differences file (diff_rev2.pdf) where these corrections are highlighted in color (additional text in blue and deleted text in red color).

Reviewer #1 (Remarks to the Author):

As I stated in my first report, this is a novel work which proposed a feasible solution for determination of (partial) configurational entropy of glass-forming system via X-ray pair correlation function. The strategy allows comprehensive discussions on important issues in glass physics, e.g., the aging/rejuvenation, relaxation, glass transition as well as atomic-scale mechanism for structural arrangement. All of which can be correlated with macroscopic calorimetric measurement. The scientific merit has justified publication in Nature Communications. It will definitely bring extensive attentions from the community, not only from experimentalists, but also those researchers in the computer simulations.

This is also a very good revision. The revised manuscript is now much solid, clear and errorless. All my concerns have been appropriately addressed. The explanations are reasonable and clear to me. Indeed it is a good demonstration of the peer review system. The revision is accepted.

With these, I would like to recommend publication of this revised manuscript as is.

We would like to thank the reviewer for devoting his time and interest to our work and for the appreciation. The reviewers contribution was very valuable.

Reviewer #2 (Remarks to the Author):

I thank the authors for carefully reading my comments and correcting the manuscript where appropriate. I believe that the discussion in section III C is now better structured and more informative.

I still have one concern about this manuscript. In my opinion there is still an issue with the discussion about the order of the β -transition that was already highlighted in my first review. In an ideal first-order phase transition one would expect to see a discontinuous jump in the entropy and a clear peak in the heat flow. This is of course not observed in Fig 3 b and instead there is a relatively broad peak from $T/T_g \sim 0.85$ to $T/T_g \sim 1.05$ for the HPT at RT sample. I agree that this is compatible with a phase transformation with first-order characteristics. My only concern is that it is localized completely in the wing and α - regions. **Why are the authors ascribing the first order characteristic to the β -transformation if the peak is located in the wing and α -regions?**

The observation of the reviewer is correct. The peak starts in the color scheme in the β -regime. At first sight it seems that there are 2 main overlapping processes. A smaller one in the β - regime and a larger one located mostly in the wing area of the color scheme extending to the α -regime. These peaks indicate first order characteristics both for the β - transition (small peak $T/T_g=0.65$ to 0.85) and the excess wing (larger peak $T/T_g=0.85$ to 1.05) and these peaks are smeared out.

With respect to the reviewers question we have 4 comments/explanations

- i) We measure the configurational entropy of the X-ray pair correlation. We observe changes in Figure 3 related to this quantity starting in the range of the β -transition as derived by DMA in torsion mode with 60 Hz with 10 K/min heating rate. The onset temperatures and the width of the transitions in DMA depend on the heating rate and on the actuation frequencies. For example, at 1 Hz the β -peak extends to higher temperatures for the as-cast state and couples with the wing in the HPT deformed sample (Figure 5 (a) of the manuscript). For a final differentiation of the different overlapping processes more research is still necessary. For representation reasons the 60 Hz curve was chosen as reference for the color scheme as the transitions are more clearly separable than at lower frequencies. In the XRD experiment the heating rate was also 10 K/s.
- ii) By evaluation of the radial distribution function (RDF) one can attribute first maxima to the main bonding of Cu-Zr, Cu-Cu and Zr-Zr [the procedure is laid out in reference 44 of the manuscript]. The fitting of these peaks indicate a higher mobility of the Cu species starting from $0.65 T_g$ but getting stronger at about $0.85 T_g$. The evaluated equivalent entropy S_{eq} is very sensitive to subtle changes in chemistry. This sensitivity could potentially enhance the signal change in the excess wing regime. We have included a diagram comparing the bond length change for the HPT deformed cases (77K and RT) for Cu-Zr (Fig. 1) and Zr-Zr (Fig 2) for the reviewer's attention. A red arrows indicates how the average bond length are changing.

Fig1: Change in Cu-Zr bond length as derived from RDF peak fitting.

Fig1: Change in Zr-Zr bond length as derived from RDF peak fitting.

- iii) Yu et al indicate in a study using MD simulations [reference 39 of the manuscript] that the excess wing stems from the same origin i.e. string like cooperative jumps, however with a larger amount. In the case of the excess wing the coupling of the α - and β -modes allows for

an even stronger relaxation of free volume due to the stronger cage breaking of the atoms. As stated in ii) the mobility of chemical species (i.e. Cu) may also change specifically.

- iv) In case that the α -transition is involving larger extents of random first order transitions at the very local scale (as the random first order transition theory RFOT suggests) our method using correlations at the atomic scale would be rather sensitive to such events. This could explain why the peak extends to the α -regime.

To conclude we can say that novel experimental and theoretical approaches often raise more questions than they can initially answer. For future synchrotron work we plan to perform entropy related studies with a focus on the partial correlations and with different alloy systems where we may have stronger or weaker β - relaxations. With these experimental results at hand we hope that we can systematically investigate the complex interactions of structure and dynamics in metallic and other glasses in correlation with calorimetry (DSC) and mechanical spectroscopy (DMA).

If the authors have a reasonable answer to this question, I think the paper should be published in its current form.

We hope that our comments above can answer the reviewer's questions. In order to highlight that the peak starts in the β -regime we have slightly adapted Figure 3 of the manuscript.

Fig3: Adapted Figure 3 of the manuscript.

Minor points:

- 1) Fig 2. Caption. Replace "shifted by a scalar factor" by "shifted by an additive constant"
- 2) Section III A. "Baranyai and Evens ..." should be "Baranyai and Evans ..."
- 3) Section III C. "Crystallization clearly involves a sharp step and can hence be interpreted as a second order transition (Figure 2)" Please change "second order" to "first order". Of course crystallization is a first order transition.

We are very thankful to the reviewer for finding these small but important mistakes in our manuscript. We have corrected them in the manuscript.

REVIEWERS' COMMENTS

Reviewer #2 (Remarks to the Author):

The authors have answered my questions and I endorse the publication of the manuscript in its current form.